# All-Optical, Air-Coupled Ultrasonic Detection of Low-Pressure Gas Leaks and Observation of Jet Tones in the MHz Range

**DOI:** 10.3390/s23125665

**Published:** 2023-06-17

**Authors:** Kyle G. Scheuer, Ray G. DeCorby

**Affiliations:** 1Ultracoustics Technologies Inc., Sherwood Park, AB T8A 3H5, Canada; 2ECE Department, University of Alberta, 9211-116 St. NW, Edmonton, AB T6G 1H9, Canada

**Keywords:** jet tone, leak detection, air-coupled ultrasound, optical ultrasound detection, optomechanics, buckled dome microcavity

## Abstract

We used an ultrasensitive, broadband optomechanical ultrasound sensor to study the acoustic signals produced by pressurized nitrogen escaping from a variety of small syringes. Harmonically related jet tones extending into the MHz region were observed for a certain range of flow (i.e., Reynolds number), which is in qualitative agreement with historical studies on gas jets emitted from pipes and orifices of much larger dimensions. For higher turbulent flow rates, we observed broadband ultrasonic emission in the ~0–5 MHz range, which was likely limited on the upper end due to attenuation in air. These observations are made possible by the broadband, ultrasensitive response (for air-coupled ultrasound) of our optomechanical devices. Aside from being of theoretical interest, our results could have practical implications for the non-contact monitoring and detection of early-stage leaks in pressured fluid systems.

## 1. Introduction

Sounds produced by flowing liquids and gases [1] play a central role in a myriad of commonplace phenomena, including human speech [2], whistles produced by animals and engineered objects [3], and, of course, the rich sounds produced by many musical instruments (e.g., wind instruments and pipe organs [4,5]). In spite of their ‘everyday’ nature, the physics of flow-induced acoustics is quite complex [6] (and ‘fundamentally non-linear’ [1]), such that exact theoretical treatments (even for relatively simple geometries) are not routinely possible.

Nevertheless, the general features of flow-derived sound are well understood. Typically, acoustic signals arise due to turbulent conditions that correlate with vibrations (i.e., pressure waves) of the flow medium itself [3]. If appropriate feedback is present, periodically spaced vortexes can form in the turbulent flow and give rise to stable oscillations at resonant frequencies corresponding to the generation of ‘flow tones’, or, in more common terms, ‘whistling’. For a given geometry, periodic vortex shedding and associated discrete tones typically arise for certain ranges of flow velocity [1]. A practical application of this phenomenon is the so-called vortex flow meter [7], in which the vortex shedding is induced via an engineered obstruction (i.e., a ‘bluff body’ [1]) placed in the flow path, and flow rates are extracted from measurement of the vortex-shedding frequencies.

In the present work, we describe a detailed experimental study of ultrasound produced by nitrogen gas jets emitted from a variety of syringes. Furthermore, we show that our observations are consistent with historical studies on gas jets emitted from much larger ‘pipes’ [8,9,10,11,12,13,14,15], albeit scaled to significantly higher frequencies in the present case. The results illustrate some unique capabilities of our recently reported [16] optomechanical ultrasound sensors, in particular their ultrasensitive and omnidirectional response to air-coupled ultrasound extending over a bandwidth of several MHz. Implications for practical applications, such as leak detection and metering of small-scale, high-pressure flows, are discussed.

The sensor we employ in this study is based on a buckled-microcavity Fabry–Perot resonator. We previously conducted several studies on the optical properties of these devices, including their suitability for quantum electrodynamics due to the high finesse (~10^3^–10^4^) and small mode volumes (as low as ~1.5 λ^3^) routinely obtained [17,18,19], their applications in microfluidics as open-access cavities for liquids [20], and their ability to be fabricated with large birefringence and polarization mode non-degeneracy [21]. More recently, we demonstrated that our devices function as extremely sensitive optomechanical sensors for both static pressure differentials [22] and ultrasonic signals in air and water [16,23]. Notably, the ultrasonic force sensitivity is one or more orders of magnitude lower than other air-coupled ultrasound sensors, with a bandwidth spanning several MHz [24,25,26]. These properties prompted us to investigate the feasibility of using our devices for gas leak detection in industrial applications.

## 2. Materials and Methods

The sensor presented herein is based on a buckled-dome microcavity, where two Bragg mirrors, one planar and one concave, are separated by a partially evacuated and sealed cavity. Our previous work describes the buckled microcavity fabrication process in detail [16,23]. Nevertheless, we also provide an overview here. Briefly, a 3.5 period Si/SiO_2_ Bragg stack centered at 1600 nm and terminated with Si was deposited on a quartz substrate via plasma-enhanced chemical vapor deposition (PECVD). Photolithography was performed with AZ1512 resist to pattern circular anti-features with a diameter of 100 µm. A ~15 nm fluorocarbon layer was then deposited, and lift-off was performed, leaving behind circular fluorocarbon pads that, ultimately, determined the dimensions of the devices. A second, identical Bragg mirror was deposited directly on top of the fluorocarbon layer and exposed bottom mirror. The substrate was then heated on a hotplate to induce bucking at the fluorocarbon sites, resulting in the formation of half-symmetric Fabry–Perot microcavities. Our previous work utilized a bulky optical table setup to conduct optical and ultrasonic measurements with our devices. This setup is not ideal for industrial applications, since rigid assembly is required to maintain optical alignment in extreme environmental conditions. To address these shortcomings, we designed and assembled a standalone probe unit using off-the-shelf optical components from Oz Optics and Thorlabs. A substrate containing our fabricated devices was affixed to threaded spacer (Thorlabs), which was then attached to a pigtail-style fiber focuser (Oz Optics). The fiber focuser was then connected to an optical circulator (Thorlabs) for interrogation and readout. A full description of the optical properties of our buckled dome microcavity devices is provided in a previous work [16], along with a detailed description of the optical interrogation and readout scheme. For the set of measurements presented herein, we used ~1 mW of optical power at ~1505 nm to couple to the dome mode and read out to the photodetector (Resolved Instruments). The photodetector was set to an 80 MHz sampling rate, and each measurement was averaged over 300 samples. No smoothing or additional post-processing was performed.

The gas system consisted of a nominally 2500 PSI N_2_ tank (Linde) initially regulated to 40 PSI. The regulator was connected to a toggle valve that was normally open during experiments, and then to a needle valve (Swagelok) and pressure gauge (Baker Instruments), providing a pressure resolution of 0.1 PSI over the range of interest. The output of the pressure gauge was connected to the needle assembly under test. Needles of various gauges (Becton Dickenson PrecisionGlide, with dimensions shown in Table 1) were attached to syringes, which were then directly attached to a section of gas line tubing. A fitting was attached to the other end of the tubing, allowing each needle assembly to be easily attached or removed from the rest of the system. All measurements were performed in ambient laboratory conditions without any specialized acoustic treatment. Figure 1 shows a schematic representation of the gas handling system, along with a photograph of the probe/needle measurement configuration used throughout this manuscript. A photograph showing the needle assemblies is available in the supplementary information.

## 3. Results

### 3.1. Observation of Jet Tones in the MHz Frequency Range

Controllable pressures were achieved by first opening the toggle valve to allow the system to pressurize up to the supply side of the needle valve. The needle valve was subsequently adjusted while monitoring the pressure gauge, resulting in a stable leak through the needle orifice. Individual power spectral density (PSD) plots for the 30-gauge needle at different pressures are shown in Figure 2, along with a colormap that characterizes the PSD as a function of pressure. At low pressures, only low amplitude features that were invariant with pressure and characteristic to the gas handling system were observed. These features were also observed for the other needle gauges, as well as without a needle assembly present, as shown in the Supplementary Information. At ~8 PSI, jet tones that were evenly spaced in frequency began to emerge. Consistent with observations from Anderson [11], the spacing of these features is not initially representative of the fundamental tone, as the fundamental is often neither the highest amplitude nor the first-appearing resonance. The true fundamental frequency spacing begins to appear near 11 PSI, and is on the order of 60 KHz for the 30-gauge needle. These jet tones are positively correlated to pressure and continue to increase with pressure until eventually combining with the noise floor around 14 PSI, where the spectrum is dominated by white noise that extends into the MHz range. This rising noise floor can also be observed through comparing the individual PSD plots shown in Figure 2a,b.

In addition to the 30-gauge needle, we performed similar characterization using both 22- and 26-gauge needles. Representative plots for each needle are shown in Figure 3, demonstrating frequency spacing that is a function of both the pressure and the dimensions of the needle. In general, the smaller the orifice diameter (i.e., higher needle gauge), the greater the spacing between jet tone harmonics and the higher they persist in frequency. As detailed in Section 4, our datasets are entirely consistent with historical observations [10,14] of acoustic signals emitted by gas jets emanating from pipe-like orifices.

### 3.2. Broadband Leak Detection

We now turn our attention to the characterization of high-pressure-differential signals where jet tones were not typically observed. Figure 4 shows the acoustic content of the 30-gauge needle in the pressure range 15.0–30.0 PSI. We observed a white noise contribution at high pressures, where the amplitude across the range 0~5 MHz was positively correlated with pressure. While the spectral content could be viewed as white noise in a flow rate sensing context, we note that the PSD is not entirely featureless, and could represent many densely spaced resonances. Regardless, the presence of spectral content in the MHz region, being orders of magnitude above the noise floor, clearly demonstrates the potential of our sensor for industrial applications, particularly in noisy settings where analysis in lower frequencies might not be possible.

We found that while PSD generally increased across the frequency range 0~5 MHz, higher frequencies seemed to be particularly sensitive to pressure. The inset of Figure 4 shows how the PSD evolves with pressure at three discrete frequencies (0.5 MHz, 1.0 MHz, and 1.5 MHz). The power measured at a constant location is proportional to the acoustic power emitted by the source (i.e., the gas jet) [27]. It is also the case that the power emitted by the source is proportional to the mass flow rate of the gas, which scales with the square root of pressure. These quantities can be related to the sound pressure level (*SPL*) using the expression:*SPL* ∝ 10 log(*W*/10^−12^) ∝ log(*ṁRT*/*M*).(1)

Here, *W* is the sound power level at the source, *ṁ* is the mass flow rate of the jet, *R* is the gas constant, *T* is the temperature, and *M* is the molecular weight [27,28]. We plotted the measured power at three distinct frequencies against the square root of the pressure differential applied to the needle. A linear fit was applied for each frequency, revealing excellent agreement (R^2^ > 0.99), though there was increasing deviation at higher frequencies. We speculate that this result could be explained by a combination of attenuation in air and the possibility that higher pressures possess higher frequency content. We also recognize that understanding jet noise associated with highly turbulent flow is complicated in its own right, and that numerous theories were previously proposed [29,30,31,32,33].

We also observed additional non-linear contributions from our sensor at sufficiently high pressures in the form of a higher-order resonance feature near 4.8 MHz in some cases. This feature was present regardless of the needle used, as shown in Figure 5. We attribute this result to incoming pressure waves causing deflections that are a significant fraction of the linewidth of the optical resonance. In such cases, the relationship between pressure and optical power becomes non-linear, and harmonics of the natural vibrational modes of the dome appear in the noise spectrum (e.g., the feature at 4.8 MHz is a second-order harmonic of the dome fundamental vibrational resonance at 2.4 MHz). Figure 5 also illustrates that the broadband frequency content associated with higher pressures is not specific to the 30-gauge needle primarily studied in this work; rather, it is present regardless of the orifice size used. Additionally, the dynamic range between 0 and 20 PSI spans several orders of magnitude in all cases.

### 3.3. Omnidirectional Detection

The comparatively small active area of our sensors (~100 µm), combined with the nature of the buckled microcavity structure, provides inherent omnidirectionality [16]. To demonstrate this fact, we varied the lateral distance between the needle and sensor, while keeping the angle between them fixed at 0°, as shown in Figure 6c, as opposed to the 90° configuration used in previous measurements (Figure 1) The PSD plots in Figure 6a,b show results for two different lateral distances using the 30-gauge needle. We found that even in this extreme configuration, the spectral content was still visible in the <500 MHz region, above which air attenuation is suspected to be the limiting factor.

We also note that this distance is not representative of the ultimate device performance we project to be possible. The primary performance-limiting factor for our probe was the coupling between the interrogation laser and device, and better performance is anticipated with future iterations. This aspect could be addressed by designing a custom probe assembly with the ability to make small adjustments to correct for micron-scale misalignment. Aside from this issue, future work will involve further investigating the impact of the mirror layer structure and device size on the optomechanical performance.

## 4. Discussion

As mentioned, our observations are consistent with the theoretical framework developed for gas jets emitted from pipe-like orifices. Here, the syringe needle itself plays the role of the pipe, and a gas flow through this needle is driven by a pressure differential between the internal body of the syringe and the external lab environment. Regimes of behavior can be understood using the well-known (and dimensionless) Reynolds (*Re*) and Strouhal (*St*) numbers. Here, *Re* = *ρvd*/*μ* and *St* = *fd*/*v*, where *ρ* and *μ* are the density and dynamic viscosity of the gas, *v* is flow velocity, *d* is a characteristic dimension (approximated by the inner diameter of the needle here), and *f* is the ‘vortex shedding’ frequency. In many flow problems, *St* is approximately constant over a wide range of *Re*, implying that the observed vortex-shedding frequencies (and associated acoustic emissions) will scale as *f*~*v*/*d*. Thus, for the very small values of *d* studied here, we expect the acoustic noise and jet tones to extend to much higher frequencies than could be detected using conventional microphones in earlier studies [8,9,10,11,12,13,14], but which are well within the capabilities of our broadband ultrasound sensors.

For the pressurized syringe, high-level details of the flow properties and acoustic signals emitted can be understood as follows [9,10,12,15]:i.Pressurized gas flows into the needle through an orifice termed the ‘vena contracta’ [12], which is an effective aperture of diameter *δ*, being slightly smaller than the inner diameter of the needle (e.g., *δ*~0.63 *d*). This orifice represents the primary obstruction in the flow path and is, thus, the appropriate characteristic dimension to use in calculation of *Re* and *St*.ii.For low *Re*, flow in the syringe needle is laminar, and negligible acoustic emissions are observed. Above some critical value of *Re*, however, eddies are expected to form near the needle entrance, resulting in a vortex trail forming along the inside walls of the ‘pipe’ and flowing in the downstream direction [10]. Within a certain range of *Re*, this vortex trail is approximately periodic, resulting in the emission of harmonically related jet tones at the exit aperture of the needle. Notably, both harmonics and sub-harmonics of the vortex-shedding frequency can be observed in the acoustic spectrum within this ‘jet tone’ regime, as observed and explained by Anderson [9].iii.For sufficiently high Re, the vortex formation becomes increasingly chaotic, and the flow becomes increasingly turbulent. In this regime, jet tones are subsumed into a broad background of ‘white’ noise, and the power spectral density of this ‘turbulent noise’ continues to increase as the flow (i.e., *Re*) is increased.

The results reported in Section 3 (and the Appendix A) are in line with these expectations. Using the relationship *v =* (2∆*P*/*ρ*)^1/2^ [5], where ∆*P* is the pressure differential across the syringe ‘pipe’, the curves plotted in Figure 3d verify the expected scaling *f*~*v*/*d* discussed above. It should be noted that it is the fundamental jet tone frequency that is plotted. Since the tones observed at lower pressure spacings do not necessarily exhibit the full spectrum of harmonics [11], full colormaps for each needle were collected (and are available in the Supplementary Information), making it easier to identify the harmonic eigenfrequencies. However, it was somewhat difficult to extract the fundamental frequency for the 22-gauge needle due to the narrow pressure range where jet tones were observed as well as their comparatively low amplitude. We fit the data from each needle to a linear equation and found the slopes to be ~0.063 KHz/KHz, ~0.070 KHz/KHz, and ~0.043 KHz/KHz for the 22-gauge, 26-gauge, and 30-gauge needles, respectively (R^2^ > 0.99 in all cases). It is interesting, though perhaps expected, that all three needles seem to exhibit similar slopes when the dependence on pressure and orifice diameter were taken into account. We suggest that the variation in slope observed might result from the fact that each gauge of needle has a different length, and that the characteristic dimension has some dependence on both *d* and *L* [9,11]. Moreover, the manufacturer does not specify a tolerance on any needle dimension, and we did not account for the needle bevel. Our data seem to suggest the absence of a jet tone in the case of a pressure differential below some threshold or for an infinitely large orifice diameter (in the form of an origin crossing), which is again consistent with Anderson’s observations [9]. Anderson performed a similar experiment with orifice plates affixed to a tube of length *L* (where *L* was the characteristic dimension of the system) and noted nearly identical slopes. Another curiosity is that the slope for each needle deviated near the upper pressure range where jet tones were observed, at least in the case of the 22- and 26-gauge needles. We attribute this result to changes in the nature of vortex formation at high *Re* values [12].

Each needle was found to exhibit a distinct range of pressures where jet tones, and turbulent flow in general, were observed. This finding suggests the possibility of estimating the size of a pinhole leak in an industrial setting if the pressure of the system is known. The diameter of the orifice could be estimated using either the jet tone spacing or the amplitude of the white noise at a given distance. The ability to measure fluid leaks has wide appeal within the oil and gas industry, and significant effort was previously expended in developing early detection systems based on ultrasonic technology [34,35,36,37,38]. However, many of these systems are limited, at least in part, by their bandwidth and sensitivity. We believe that the ability to measure leaks from sub-millimeter holes at low pressures far into the MHz-frequency-range will be of particular interest for hazardous or explosive gases, where electronic components cannot be placed in close proximity. Additionally, many sources of ambient noise lie well below the MHz-range signals detected in the present work, which is a significant potential benefit of the broadband capabilities of our sensor. In future work, we hope to target the direct measurement of gas leaks in an industrial setting.

One additional intriguing possibility suggested by our results is that a small needle emitting a controlled gas jet could represent one way of generating a tunable acoustic frequency comb, especially since the pressures used here should be accessible to any space that already utilizes floating optical tables. This finding could represent a partial setup for photo-acoustic comb spectroscopy without requiring expensive acousto- or electro-optic modulators [39,40].

## 5. Conclusions

In summary, we have made three distinct contributions. First, as a compact probe for all-optical detection of MHz-frequency range, air-coupled ultrasound was constructed. Second, we generated MHz-range jet tones by passing pressurized nitrogen gas through a collection of small syringes and showed that previously established theory can be extended to this range. Finally, we showed that high-pressure gas leaks contain frequency content that extends far into the MHz range, lying orders of magnitude above the noise floor of our devices. Additionally, gas leaks can be sensed both off-axis and off-position. Our buckled microcavity-based devices function as uniquely enabling leak sensors due to their sensitivity, bandwidth, and omnidirectionality.

## Figures and Tables

**Figure 1 sensors-23-05665-f001:**
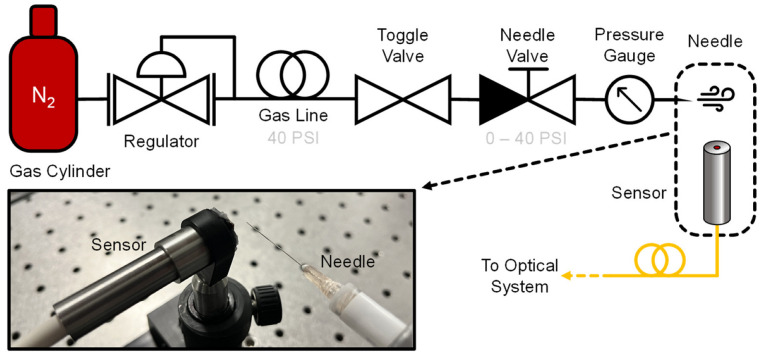
Experimental scheme showing gas handling setup. The setup served to both generate jet tones characteristic to the needle and simulate a controlled gas leak at an arbitrary pressure in range 0–40 PSI. The photograph shows the measurement configuration used throughout the manuscript, unless otherwise specified.

**Figure 2 sensors-23-05665-f002:**
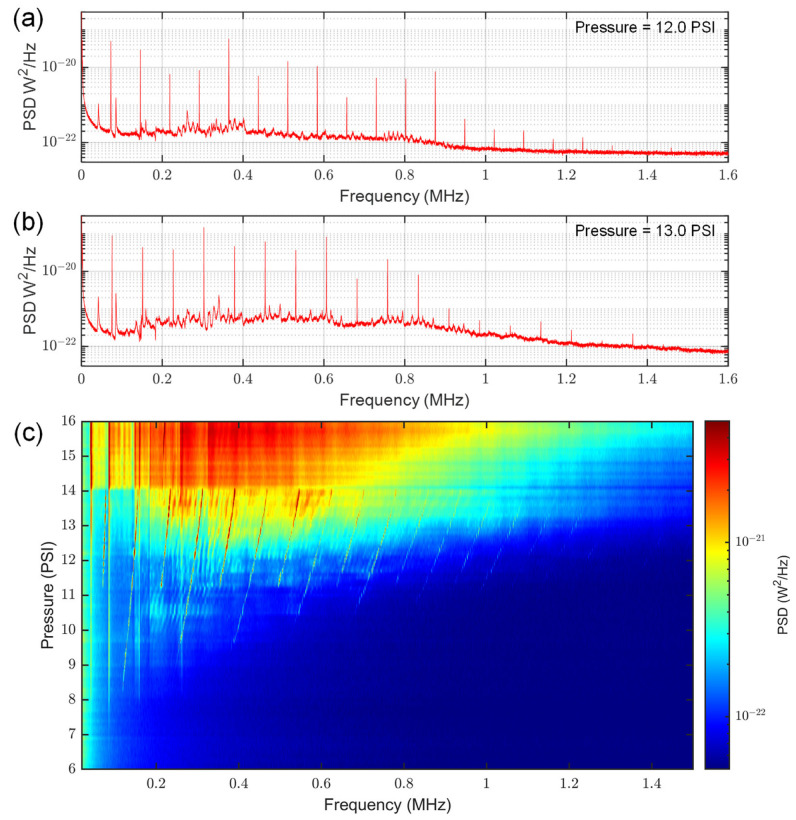
Generation of MHz frequency jet tones with a 30-gauge needle at various pressures. The sensor was placed 1 cm from needle at a 90° angle, as shown in Figure 1. (**a**) A power spectral density plot at 12.0 PSI. (**b**) A power spectral density plot at 13.0 PSI. (**c**) A colormap characterizing jet tones as a function of pressure. Jet tones are present in the region 8~14 PSI, while other static features inherent to gas handling system are present throughout a broader pressure range.

**Figure 3 sensors-23-05665-f003:**
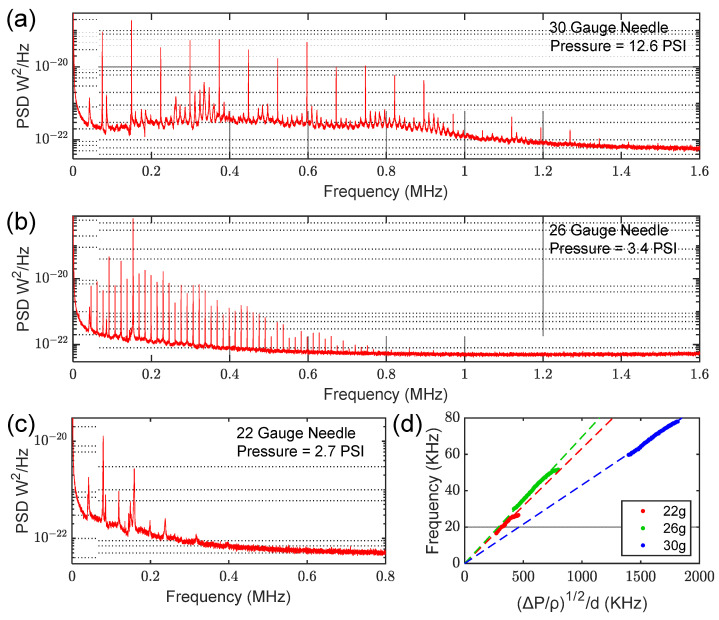
Generation of jet tones with a variety of needles, showing a clear dependence on dimensions of each needle. The sensor was placed 1 cm from needle at a 90° angle, as shown in Figure 1. (**a**) A power spectral density plot for the 30-gauge needle at 12.6 PSI. (**b**) A power spectral density plot for the 26-gauge needle at 3.4 PSI. (**c**) A power spectral density plot for the 22-gauge needle at 2.7 PSI. (**d**) A comparison of the fundamental jet tone frequency for all needle gauges as a function of pressure differential and orifice diameter.

**Figure 4 sensors-23-05665-f004:**
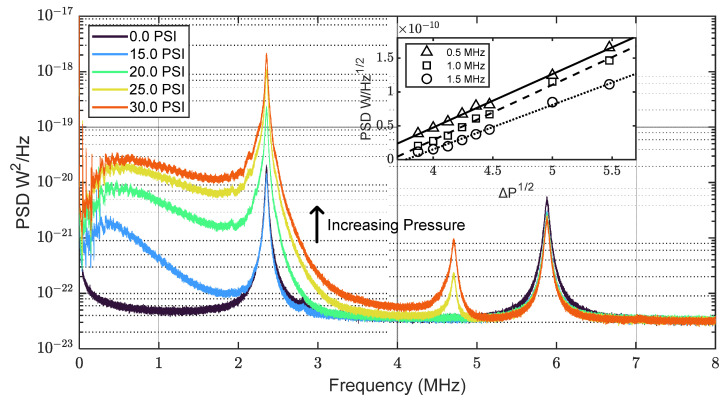
Sensing broadband frequency content of high-pressure nitrogen jets. Each trace shows measured PSD as the pressure of gas line was varied. The sensor was placed 1 cm from needle at a 90° angle, as shown in Figure 1. The plot illustrates that high pressure gas jets through small orifices possess spectral content well into the MHz frequency range, and that our sensor can detect such signals.

**Figure 5 sensors-23-05665-f005:**
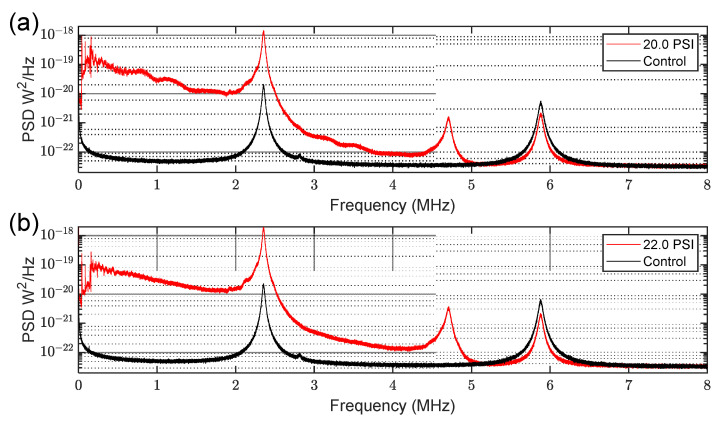
Broadband frequency content for additional needle gauges. Each needle was measured using the 90° needle–sensor configuration shown in Figure 1. A control measurement with the gas handling system depressurized is also presented for each case. (**a**) PSD plot for the 22-gauge needle at 20.0 PSI; (**b**) PSD plot for the 26-gauge needle at 22.0 PSI.

**Figure 6 sensors-23-05665-f006:**
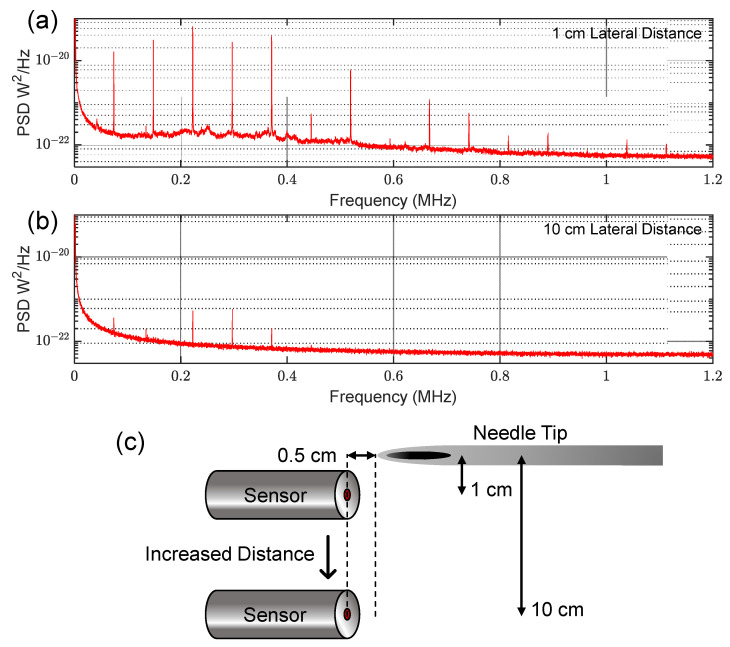
Omnidirectionality demonstration using the 30-gauge needle. PSD plots were recorded as sensor was moved laterally with respect to needle tip. The gas system was held at a constant pressure of 12.3 PSI. (**a**) PSD plot at a lateral distance of 1 cm; (**b**) PSD plot at a lateral distance of 10 cm; (**c**) Schematic showing the configuration for each measurement.

**Table 1 sensors-23-05665-t001:** Nominal dimensions of needles used in this work.

Gauge	Inner Diameter (mm)	Outer Diameter (mm)	Length (mm)
22	0.413	0.7176	38
26	0.260	0.4636	13
30	0.159	0.3112	25

## Data Availability

The data that support the findings of this study are available from the authors upon reasonable request.

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
