# Peer review of "All-Optical, Air-Coupled Ultrasonic Detection of Low-Pressure Gas Leaks and Observation of Jet Tones in the MHz Range"

_sensors, 2023, doi:10.3390/s23125665_

Round 1

Reviewer 1 Report

This is a very interesting paper. The research ideas and content are both innovative. In this paper,  a detailed experimental study of ultrasound produced by nitrogen gas jets emitting from a variety of syringes was conducted, A series of interesting experimental results have also been analyzed in detail. At the application level, the research results presented in this paper are of positive significance for non-contact early-stage leak detection in pressured fluid systems. For the aforementioned reasons, I recommend that this article be published in this journal after making some minor revisions.

(1) the paper introduction can be enhanced, Especially, the author’s own previous research foundation can be appropriately introduced in the introduction, which would be helpful for readers to understand the current status of the entire study.

(2)The language in this article needs to be strengthened, as some expressions are colloquial.

The language side need to be improved. Please revise.

Reviewer 2 Report

The paper presents the experimental results, with adequate discussion, for the test stand dedicated to investigate feasibility of detection of low-pressure gas leaks making use the probe constructed by the Authors. The probe and the test stand was design to examine the air-coupled ultrasonic measurements for the jet tones generated in the 0-5 MHz frequency range. Nitrogen gas jets emission using syringes was considered in the study. Specifically, acoustic signals generated by the escaping pressurized gas were analyzed. The work presents interesting concluding regarding relationship between the jet tones and characteristic parameters of the induced gas flow. It is worth mentioning that the conducted research should be considered of high practical importance for detection of early-stage leaks in fluid systems. Additionally, the current research makes a reference to the Authors’ previous work (buckled microcavity fabrication process and preliminary optical and ultrasonic measurements). In summary, the work states for a comprehensive analysis conducted for various pressures, needle’s diameter and examined frequency. In the reviewer’s opinion the work is worth to be published after the below-stated comments are addressed by the Authors and adequate comments added to the manuscript.

Is it possible to qualitatively/quantitatively scale (project) the observed phenomena and measured parameters to other cases with different dimensions of the pipes, large/small orifices and other types of leakages? I think a short comment on that in the paper would be interesting.

The Authors claims to design “ultrasensitive” sensor. A comment on the comparison with other available ones would be more informative for the reader.

How to detect two or more gas leaks in the same region (possibly of various severity- orifices diameters and pressure) using the proposed approach?

What is (would be) the influence of ‘real-life’ environmental conditions, including temperature and humidity variation over time?

There is a very interesting and crucial conclusion stated that the distance between the needle and sensor is not representative of the ultimate device performance. What are, in the Authors’ opinion, the most influential factors deciding about the resultant performance of the device?

What if the examined installation additionally undergoes vibration or, what seem seven worse, structural sound originated from its regular exploitation? How to prevent from that undesired interference?

Is the shape of opening (different from a round one) critical for both the presented measuring technique performance and sensed damaged indices?

It is seen potential practical application of the proposed measurement technique (the ability to measure fluid leaks and developing early detection systems). The question, however, is how the viscosity and mass density influence the performance and sensitivity of the device?

Minor issues, flaws:

Row 74: I would probably substitute the expression “300 x averaging” with some more formal one

76: Would’t “scheme” be better  than “schematic”?

137: “We now turn our attention the characterization” -> “We now turn our attention towards/to the characterization”
